# Megathrust reflectivity reveals the updip limit of the 2014 Iquique earthquake rupture

Bo Ma [1✉], Jacob Geersen[1,2], Dietrich Lange [1], Dirk Klaeschen[1], Ingo Grevemeyer [1], Eduardo Contreras-Reyes[3], Florian Petersen [1], Michael Riedel [1], Yueyang Xia [1✉], Anne M. Tréhu[4] & Heidrun Kopp [1,2]

The updip limit of seismic rupture during a megathrust earthquake exerts a major control on the size of the resulting tsunami. Offshore Northern Chile, the 2014 Mw 8.1 Iquique earthquake ruptured the plate boundary between 19.5° and 21°S. Rupture terminated under the mid-continental slope and did not propagate updip to the trench. Here, we use state-of-the-art seismic reflection data to investigate the tectonic setting associated with the apparent updip arrest of rupture propagation at 15 km depth during the Iquique earthquake. We document a spatial correspondence between the rupture area and the seismic reflectivity of the plate boundary. North and updip of the rupture area, a coherent, highly reflective plate boundary indicates excess fluid pressure, which may prevent the accumulation of elastic strain. In contrast, the rupture area is characterized by the absence of plate boundary reflectivity, which suggests low fluid pressure that results in stress accumulation and thus controls the extent of earthquake rupture. Generalizing these results, seismic reflection data can provide insights into the physical state of the shallow plate boundary and help to assess the potential for future shallow rupture in the absence of direct measurements of interplate deformation from most outermost forearc slopes.

[1] GEOMAR Helmholtz Centre for Ocean Research Kiel, Kiel, Germany. [2] Institute of Geosciences, Kiel University, Kiel, Germany. [3] Departamento de Geofísica, Facultad de Ciencias Físicas y Matemáticas, Universidad de Chile, Santiago, Chile. [4] Oregon State University, College of Earth, Ocean, and Atmospheric Sciences, Corvallis, USA. ✉email: bma@geomar.de; yxia@geomar.de

Megathrust earthquakes result from the sudden failure of the plate boundary in a region where elastic strain has accumulated prior to the event. Fluid pressure is a critical parameter that determines the physical nature of the megathrust and therefore exerts a main control on where and how seismic moment is released[1,2]. Pore fluid pressure in excess of hydrostatic pressure diminishes fault strength[1] and enables a wide spectrum of transient, predominantly slow, earthquake phenomena especially along shallow subduction zone plate boundaries[3]. The key processes that release water in shallow subduction zones, and thus control fluid pressure, are compaction dewatering and clay and opal dehydration reactions in subducting sediments and the upper oceanic basement[2,4–6]. These processes take place under low temperatures and low confining pressures[7,8], so excess pore pressures are expected in areas of low overburden, such as the shallow plate-boundary. Seismic reflection data are sensitive to the high acoustic impedance contrast generated by fluids and thus seismic reflection studies have proven powerful to investigate spatial variations in plate boundary fluid pressure, at least in a qualitative manner[5,9–15]. For the erosive convergent margin of Central America, ref. [5] suggested a high fluid content along the shallow aseismic section of the plate boundary (also compare to refs. [15,16]) in a temperature regime where compaction dewatering and clay-mineral diagenetic reactions are expected to release most of the water in the subducting sediments[2]. The authors further described a rapid decrease in fluid content where temperatures exceed ~150 °C, which corresponds to the transition from aseismic sliding at shallow depth to stick-slip sliding in the seismogenic zone.

On 1 April 2014, the Mw 8.1 Iquique earthquake ruptured the plate boundary between 19.5° and 21°S along the erosive continental margin of Northern Chile[17–20]. Seismic rupture did not break updip to the trench but terminated under the mid-continental slope (Fig. 1). Aftershocks of the 2014 Iquique earthquake concentrated around the updip limit of seismic rupture with little activity towards the trench[21,22].

In this work, we use high-resolution multichannel seismic reflection profiles in a grid layout covering the 2014 Iquique mainshock and aftershock region as well as the surrounding forearc not affected by seismic rupture (Fig. 1). The seismic reflectivity variations along the plate boundary elucidate the spatial variation in megathrust fluid pressure.

## Results

### Correspondence between plate boundary reflectivity and rupture area of the 2014 Iquique earthquake.

Pre-stack depth migrated seismic reflection profiles with a total length of 912 km cover the Northern Chilean marine forearc in the region that ruptured during the 2014 Iquique earthquake and the adjacent un-ruptured forearc (Fig. 1). The shallow plate boundary underneath the lower continental slope is visible as a prominent seismic reflection on all profiles. At greater depth, the reflectivity shows a high degree of variation in the dip direction and along strike. Along the northernmost line MC04, located to the north of the 2014 Iquique rupture where only sparse aftershocks occur (Fig. 1; ref. [22]), the plate boundary is imaged as a band of strong, albeit discontinuous, reflectivity from the trench to at least 103 km landward of the deformation front (Figs. 1 and 2a, yellow lines and arrows), where it is at a depth of about 35 km. Between profile 0 and 35 km, the plate boundary is imaged with a high degree of lateral coherency from the deformation front adjacent to the trench axis to a depth of around 15 km on all profiles (Figs. 1 and 2b–d, yellow lines and arrows). South of MC04, however, plate boundary reflectivity drops suddenly and is below

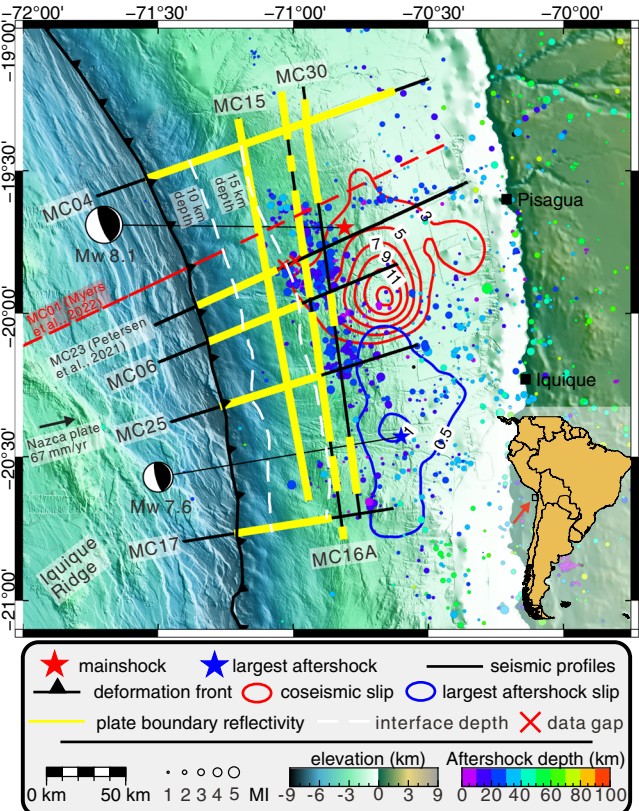

**Fig. 1 Overview map of the erosional margin of Northern Chile in the region affected by the 2014 Iquique earthquake.** The hypocenter (star) and slip contour lines (in meters) of the 2014 Iquique Mw 8.1 mainshock (red) and Mw 7.7 aftershock (blue) are from ref. [58]. Black lines indicate the locations of the seismic lines used in this study with yellow regions indicating a coherent plate boundary reflection. The location of MC23[22] has been added in the overview map, where the same color is used for the plate boundary reflectivity. The white dashed lines show the depth of the plate interface estimated from the seismic reflection data. The colored circles according to the color bar are aftershocks (December 2014 until October 2016) from the 2014 Iquique earthquake recorded by ocean bottom seismometers indicated as green and orange triangles[22]. Seafloor bathymetry from ref. [28] combined with GEBCO_2019 bathymetry (www. gebco.net), SRTM topography from ref. [59]. The convergence of the Nazca and South America plate indicated by a black arrow[60]. The location of the seismic line defining the structure of the incoming plate[40] is shown by a red dashed line.

the background noise level more than 30–35 km east of the trench.

The down-dip variations in plate boundary reflectivity are also seen on lines parallel to the trench (MC15, MC16A and MC30; Fig. 3) which image the lower continental slope with increasing distance from the deformation front (Fig. 1). The intersections of the trench-parallel lines with the dip lines provide an independent verification of the reflection character of the plate boundary at the crossing points. Along strike line MC15 which is located closest to the trench, a highly coherent plate boundary reflection is observed along the entire line at a depth of 11.5-16 km (Fig. 1, yellow line; Fig. 3a, yellow arrows). Strike line MC16A, located ~16 km farther downdip, shows intermittent high reflectivity on the plate boundary between profile kilometers 0–6, 20–33, 38–88, 94–137,145–150 and 157–175 km (Fig. 3b). Elsewhere, the reflectivity is moderate or even absent. Again, the reflectivity pattern matches the reflectivity of the dip lines, as evidenced at the cross points and intersections with MC25 and MC04

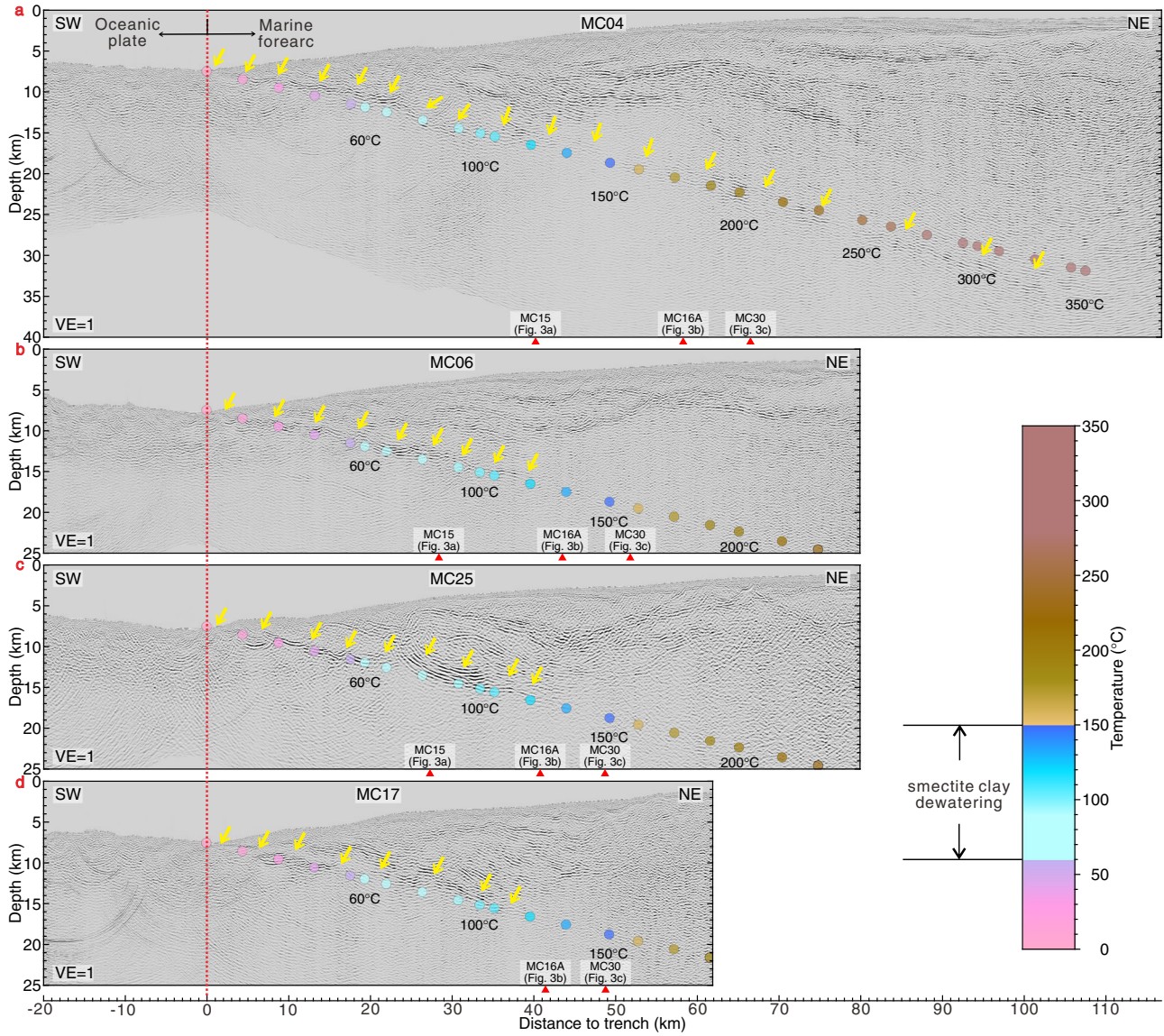

**Fig. 2 Pre-stack depth migrated section of seismic dip-lines.** Yellow arrows indicate a coherent plate boundary reflection. The vertical red dashed line denotes the location of the deformation front. The red arrows specify the intersections with the strike lines. Temperatures are shown as colored dots according to the color bar. The approximate depth range of smectite clay dehydration is based on refs. [2,7,29,32,33]. **a** seismic line MC04. **b** seismic line MC06. **c** seismic line MC25. **d** seismic line MC17.

(coherent plate boundary reflection) as well as MC06 and MC17 (weak or absent plate boundary reflection). Only ~8 km farther landward, strike line MC30 shows a completely different pattern of plate boundary reflectivity. Here the plate boundary is located at depths between 17 and 21 km (Fig. 3c). Along line MC30, the plate boundary reflection is absent or very weak, except around the intersection with MC04, which is consistent with the reflection signal on MC04 at the corresponding depth.

We use an analytical thermal model by following the approach of ref. [23] to calculate the temperature as a function of depth at the interplate fault zone (see details in the Supporting Information). Thermal constraints[24] reveal the temperature structure along the plate boundary from the trench axis downwards. The lower limit of the coherent and highly reflective plate boundary around 15 km depths corresponds to a temperature of 100–150 °C (Fig. 2). Further down-dip, plate boundary reflectivity decreases remarkably rapid in the rupture area of the 2014 Iquique earthquake and its main aftershock. Most noticeable, it also stands in sharp contrast to the region immediately north of the

rupture zone (MC04), where a moderate plate boundary reflection is observed to a depth of 35 km, corresponding to a temperature well above 300 °C.

Ref. [25] observed down-dip variations in plate boundary reflectivity over short (5–10 km scale) distances. The maximum depth extent of reflectivity in their data is consistent with our observations. The seismic data used by ref. [25] were recorded in 1995 with a significantly shorter streamer and thus lack resolution at larger depths (>16 km) compared to data from our seismic campaign and did not resolve the dramatic decrease in reflectivity in the region of the 2014 rupture.

**Impact of fluid pressure on the 2014 Iquique earthquake rupture.** A striking observation is the spatial correlation between the reflection character of the plate boundary and the rupture areas of the 2014 Iquique Mw 8.1 earthquake and the Mw 7.7 aftershock, and the aftershock distribution (Fig. 1). Updip of both rupture areas, a highly reflective plate boundary is coherently imaged on all seismic lines. Subduction zone plate boundaries

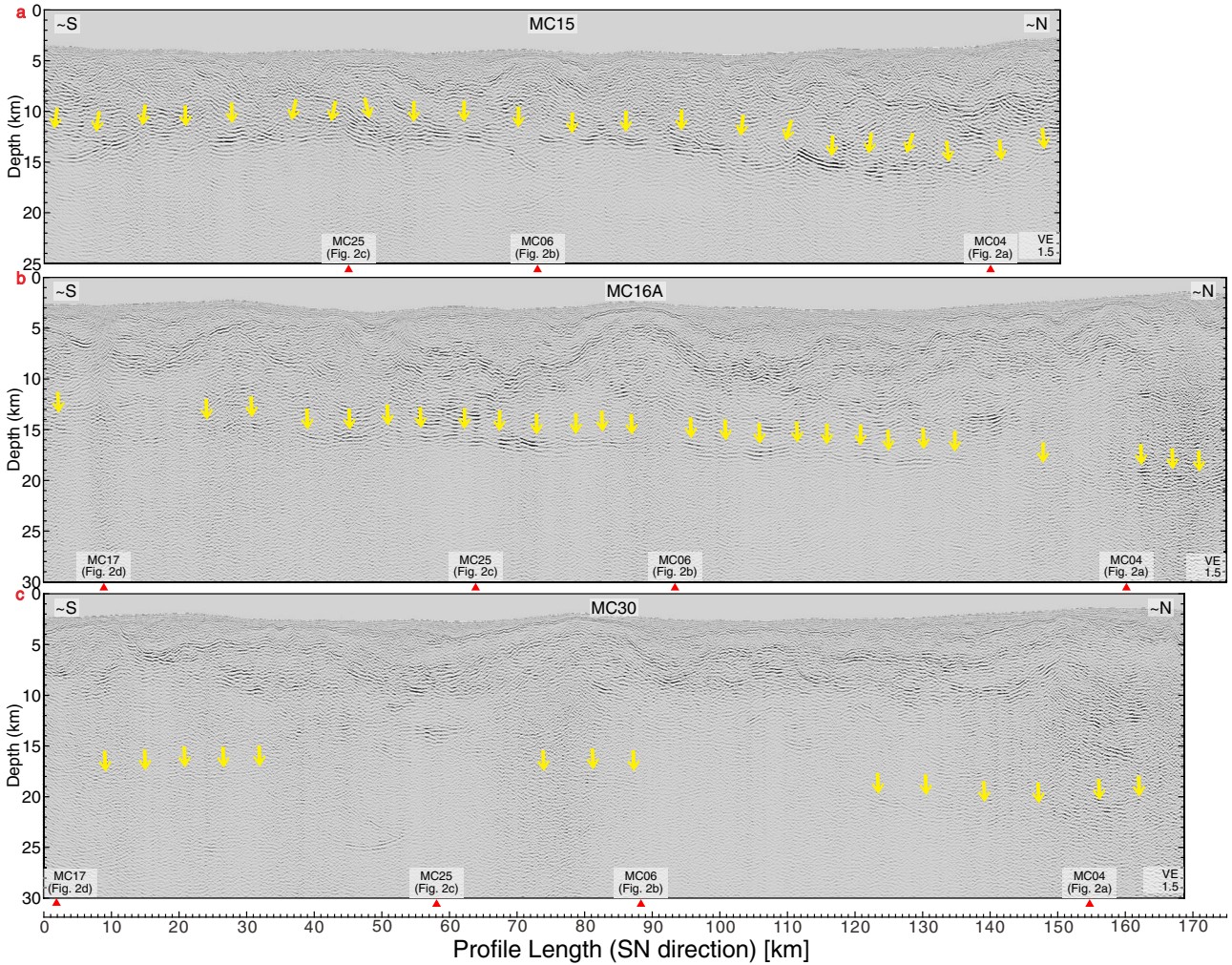

**Fig. 3 Pre-stack depth migrated sections of seismic strike-lines.** Yellow arrows indicate a coherent plate boundary reflection. The red arrows specify the intersections with the dip lines. **a** seismic line MC15. **b** seismic line MC16A. **c** seismic line MC30. All symbols as in Fig. 2.

that are imaged as coherent high reflections in seismic data are usually interpreted as high porosity and fluid rich fault zones[11,12,15]. In many subduction zones, some of which have been targeted by scientific ocean drilling (Northern Barbados, Costa Rica, Nankai Trough, Hikurangi), such coherent, high reflectivity corresponds to the shallow region of the plate boundary that does not nucleate giant earthquakes[5,9–15]. Advances in seafloor geodesy, in concert with scientific drilling of shallow plate boundaries and numerical modelling, have further proven that these regions can host a wide spectrum of slow earthquake phenomena with event durations up to some years[3,26]. For example, on the Nankai margin, recurring slow-slip events along the high-reflective shallow plate boundary, accommodate 30–55% of the plate convergence[26].

Along the erosive Central American margin, ref. [5] observed a highly reflective shallow plate boundary (also compare to refs. [15,16]). In their conceptual model, the high reflectivity is mainly caused by dehydration of sedimentary smectite clays at temperatures below 150 °C. The released fluids reduce the strength of the shallow plate boundary and migrate upwards through an upper plate dissected by large normal faults. Off Northern Chile, the shallow plate boundary is also highly reflective and the outermost part of the marine forearc is similarly fractured by long-term subduction erosion[22] and the subduction of excess lower plate topography due to the Iquique Ridge[27]. The small volume of trench fill along the Northern

Chilean margin (Fig. 2 and Suppl. Fig. 8, also compare to ref. [28]), however, makes it questionable whether subducting sediments are a significant source of fluids. If smectite clay accounts for 50% of the bulk sediment, as observed in other subduction zones with typical deep-water pelagic and hemipelagic environments of mudstones such as the Nankai Trough, Cascadia, Barbados Ridge, Costa Rica[29,30], this would correspond to 8–10 wt% of water (~15–20 vol%). Considering that the thickness of sediments that rest on the igneous oceanic basement is generally less than 200 m (and often even less than 100 m) along the Northern Chilean margin (Suppl. Fig. 8, red dots of error bar and yellow solid lines in (a)–(d)), water release from mineral dehydration within the subducting sediment must be quite limited, even when considering high smectite clay concentrations.

Recent deep-ocean drilling campaigns targeting the oceanic plates offshore Costa Rica and Nankai recovered oceanic basalts with smectite concentrations up to 40 vol%[6,31]. In both cases, the smectite clay was formed as an alteration product during the basalt interaction with sea water. Dehydration of weathered clay-bearing basalt within the uppermost oceanic basement is generally less well studied and understood compared to dehydration reactions within the subducting sediments[2,7,29,32,33], or metamorphic reactions that release fluids from the deeper sections of the oceanic crust and the upper mantle at temperatures >300 °C[34–36]. However, similar to what is happening in the subducting sediments, dehydration of the subducting

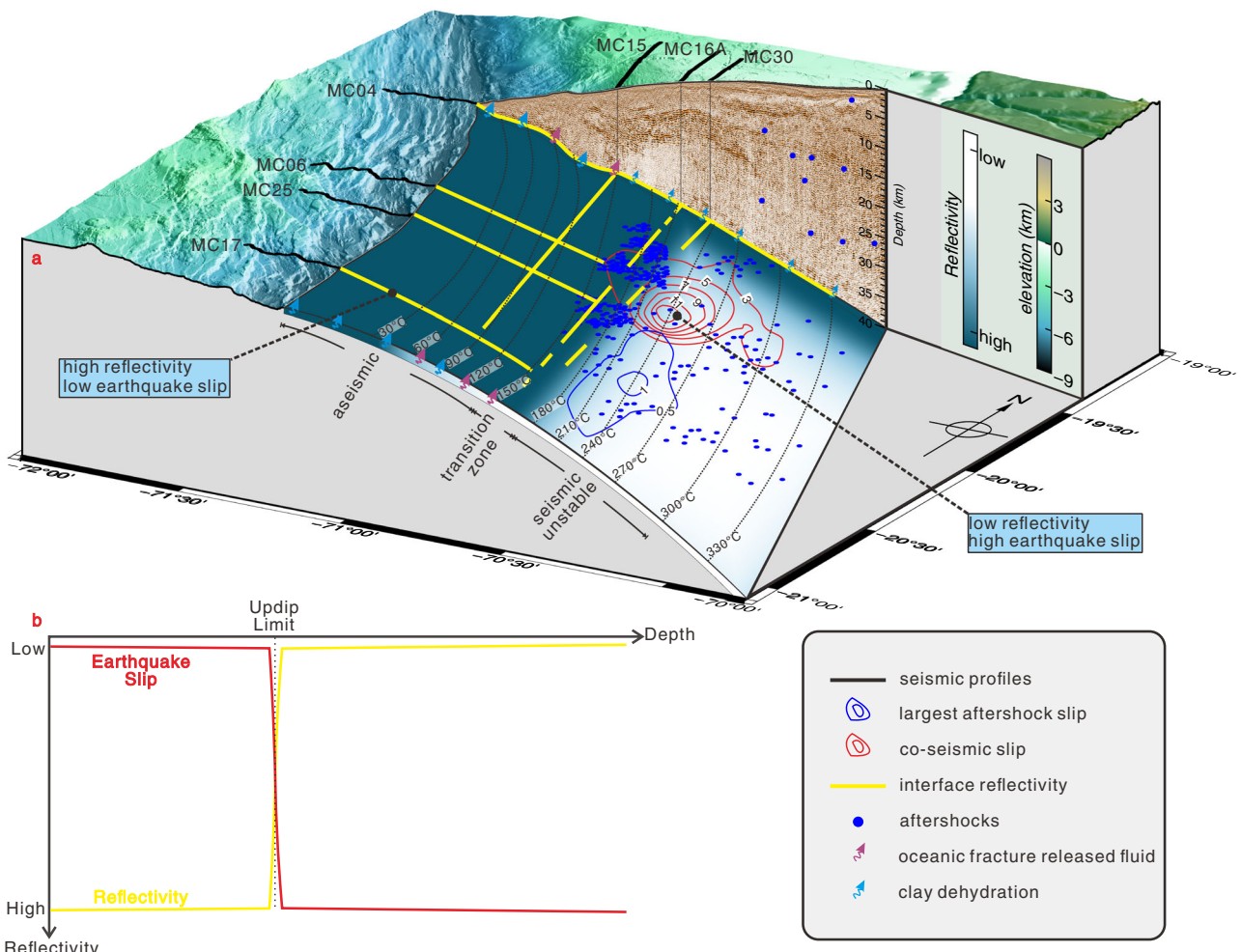

**Fig. 4 Conceptual model of the seismotectonic and hydrogeological setting in the region of the 2014 Iquique Mw 8.1 earthquake. a** Dehydration of weathered clay-bearing basalt (green arrows) and fluid release from intergranular and fracture porosity in layer 2 A (red arrows) results in a fluid rich shallow plate boundary. **b** Schematic trends in seismic reflectivity (yellow) and earthquake slip (red) with increasing depth along the plate boundary.

weathered basalt may initiate between the 60° and 150 °C isotherms[2,6,32,37,38]. For the fossil margin complex of the Shimanto Belt (southwest Japan), the smectite to chlorite conversion within the uppermost oceanic basement is discussed as a major source of water[6,39]. Furthermore, the high $v_p/v_s$ ratio derived from a recent local earthquake tomography using ocean bottom seismometers[22] is indicative of hydration of the oceanic crust underneath the lower forearc. To calculate the total amount of water that could be liberated from the scarce subducting sediments and the subducting weathered oceanic basement, one would need to know the exact composition of the sediments as well as the depth extent and degree of weathering of the clay-bearing basalts.

Although the incoming oceanic crust off northern Chile has not been drilled, recent seismic data suggest that the upper crustal velocity of the incoming plate beneath the outer rise is anomalously low, suggesting the presence of mineral hydration and/or pervasive fluid-filled cracks to a depth of ~5 km[40]. We conclude that the coherent highly reflective shallow plate boundary off Northern Chile, which behaved aseismic during the 2014 Iquique earthquake, is fluid-rich, and that the source of the fluid is intergranular and fracture porosity in seismic layer 2A[6,37,41] of the oceanic crust and mineral bound water released through dehydration of the weathered clay-bearing oceanic basalt.

This may promote reduced coupling as suggested by ref. [42] for the southern boundary of the 2010 Maule earthquake. Furthermore, the high fluid pressures and the low effective stresses may promote strain release during slow and aseismic events that occur at shorter intervals compared to large megathrust earthquakes. While observed along the Nankai[26], Hikurangi[43], and Costa Rica[44] margins, such slow and shallow earthquake phenomena have not been resolved off Northern Chile, where Global Navigation Satellite System-Acoustic (GNSS-A) seafloor observations and borehole observatories are lacking.

Farther downdip, into the seismogenic portion of the 2014 Iquique earthquake, the coherent reflectivity of the plate boundary diminishes rapidly where the plate boundary slipped in 2014 but not immediately to the north of the 2014 rupture zone (Fig. 1). For the Central American margin and elsewhere, similar observations are usually explained by a reduction in fluid pressure and/or thinning of the fault zone to a thickness that lies below the resolution of seismic reflection data[13,15]. The well-drained region farther downdip promotes the build-up of elastic strain over decadal to centennial timescales that was released during the 2014 Iquique earthquake. In particular, the three strike lines that sample the region seaward of the updip limit, the updip limit itself, and the region just below the updip limit, demonstrate the turnover from the coherent, strong reflective shallow plate

boundary that did not rupture to the non-reflective plate boundary within the 2014 Iquique earthquake rupture area. While seismic resolution is a function of energy penetration with depth, profile MC04 (Fig. 2a) documents sufficient resolution of our data set to image the plate boundary to a depth of ~35 km. While in general the reflection intensity decreases with increasing depth due to intrinsic attenuation and the viscoelastic material behavior of the subsurface[45], the very rapid and rigorous disappearance of the plate boundary reflection over a very short depth range along-strike, in conjunction with the temperature isotherms increasing to beyond 150 °C precludes a seismic imaging problem.

For the Antofagasta Mw 8.0 earthquake, ref. [46] compared upper plate seismic velocities from before and after the earthquake and suggested that seismic rupture enabled fluids to migrate upwards from the plate boundary into the upper plate. Such a process could reduce plate boundary reflectivity during the early postseismic phase and would thus provide a possible explanation for the observed low reflectivity within the 2014 Iquique rupture area. In general, there is little information on transient changes of reflectivity during a seismic cycle. Prior to the 2014 Iquique earthquake ref. [25] observed high plate boundary reflectivity down to a depth of 16 km. This indicates that the shallow plate boundary remains fluid rich throughout the seismic cycle. Since plate boundary reflectivity at the depths of the seismogenic zone is not resolved prior to the 2014 Iquique earthquake[25], we cannot conclude whether it was altered by the 2014 Iquique earthquake. However, as the updip end of seismicity, which correlates to the updip extent of the 2014 Iquique rupture, is stable in space throughout the seismic cycle, as also observed for the Sumatran[47] and the South Chilean margin[48], we presume that the reflectivity may also not change significantly over time. Although the ultimate reason for the higher plate boundary fluid content at depths beyond 15 km to the north of the 2014 Iquique earthquake remains enigmatic, our seismic reflection data clearly suggest a hydrogeological control on the updip extent and likely also the along-strike extent of seismic rupture along the erosive Northern Chilean continental margin.

**Implications for assessing the hazard of shallow earthquake rupture**. The updip extent of seismic rupture during a plate boundary earthquake exerts a major influence on the magnitude of the associated tsunami. Models using only land geodetic measurements, however, cannot resolve whether the shallowest part of a plate boundary is locked over a time period that is long enough to accumulate sufficient elastic energy to nucleate a large earthquake[49–52]. Alternatively, the shallow plate boundary may also creep at plate convergence rates or release energy during frequent slow-slip events, low-frequency earthquakes, or episodic tremor and slip[3,53]. For the 2014 Iquique earthquake, we show that seismic reflectivity of the plate boundary is spatially related to the rupture area, with coherent, high reflectivity in the shallow aseismic regions that did not rupture and weak to absent reflectivity farther downdip within the rupture area (Fig. 4). This does not exclude the possibility that earthquake rupture arising from the downdip seismogenic zone may propagate into and through the shallow plate boundary, as it has been observed and inferred from other margins[30,54,55].

Seismic reflection data, which are sensitive to fluid-pressure variations, can help to identify the shallow, velocity strengthening part of the plate boundary that may not nucleate a large earthquake and that may accumulate and release strain in a different manner and at different timescales (intervals), compared to the non-reflective seismogenic zone located further downdip.

This knowledge may provide crucial, but often missing, information towards a comprehensive evaluation of seismic and tsunami hazard along active margins, especially close to the trench where land-based geodetic and seismological studies lack resolution and offshore geodetic data are sparse or missing, but where the hazard of tsunami and tsunamigenic earthquakes is greatest.

## Methods
**Multichannel seismic reflection data**. Seismic multichannel reflection data used in this study were acquired in 2016 during the MGL1610 cruise of *R/V Marcus G. Langseth* offshore Northern Chile[56]. Seismic signals were generated with a source of 6600 cubic inches (108.15 liters), provided by four strings of 10 air-guns each. Data were recorded with an 8 km long streamer towed by *R/V Marcus G. Langseth*. The source was towed at 12 m depth below the water surface, and the record length was 16 s. The shots were acquired using a shot interval of 125 m to avoid interference from previous shots on ocean bottom seismometer data that were being acquired simultaneously. Surface-related multiple prediction, anomalous amplitude noise attenuation and adaptive filter are effective methods to attenuate multiples in our 2D seismic data. The plate boundary could be observed as a low frequency response of the seismic record. After that, pre-stack depth migration was applied to the sections with the aim to reveal more structural details. See Supplementary Methods for more information.

**Thermal modeling**. We calculated the thermal state along the plate interface megathrust fault using an analytical solutions[23] and compared the result to a two-dimensional thermal model for northern Chile[24] that incorporates corner flow in the mantle wedge. See Supplementary Methods for detailed information on the thermal modelling.

## Data availability
The multichannel seismic raw data is archived in https://www.marine-geo.org/tools/entry/MGL1610 and https://www.rvdata.us/search/cruise/MGL1610. Aftershocks[22] can be accessed via https://doi.org/10.1594/PANGAEA.929899. Multibeam data[28] can be accessed via https://doi.org/10.1594/PANGAEA.893034. We thank K. Davenport for providing a preliminary version of an unpublished 3D $v_p$ model. Figures were generated using GMT[57].

## Code availability
The thermal model[23] with varying effective coefficient of basal friction values as MATLAB source code and thermal field file[24] are available at Zenodo: https://doi.org/10.5281/zenodo.6536567.

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

## Acknowledgements

B.M. acknowledges funding from the China Scholarship Council (grant 201706400073). We gratefully acknowledge the acquisition of the MCS data and bathymetric data during cruise MGL1610 of the *R/V Marcus G. Langseth* in 2016 (United States National Science Foundation grant OCE-1459368 to Oregon State University). E.C.R. thanks the support of the ANID/FONDECYT grant 1210101.

## Author contributions

B.M. processed the seismic data with the support of D.K. B.M., J.G., D.L. and H.K. wrote the initial draft of the paper, which was critically revised by all co-authors. E.C.R. established the thermal model. I.G. and E.C.R. contributed to the discussion of the thermal models and implications of the findings. F.P., M.R., A.T., E.C.R. participated in

cruise MGL1610 of *R/V Marcus G. Langseth*. E.C.R., A.T., M.R., and Y.X. contributed to the discussion of the seismic data. I.G., D.L., F.P. contributed to seismological topics.

## Funding

## Competing interests
The authors declare no competing interests.
