## [Peer Review File · Nature Communications]

REVIEWER COMMENTS

Reviewer #1 (Remarks to the Author):

Megathrust reflectivity reveals the updip limit of the 2014 Iquique earthquake rupture - By Ma and co-authors

The ms presents seismic reflection data acquired offshore Northern Chile in the area of the 2014 Iquique EQ. The authors analyze the reflectivity of the plate boundary down-dip and along-strike and infer that the updip-propagation of the co-seismic rupture is related to the fluid content and possibly fluid overpressure along the plate boundary.

The ms is well written and the figures are appropriate and of good quality.

My review focuses on three points, that, in my opinion, need further thinking.

The first one regards the temperature proposed for the plate boundary at different depths. The authors mention numerical models performed by Kellner (2007). I reckon this is an unpublished thesis (Ms? PhD?). Temperature is a key point in this paper and I would like to see how this modelling is done. Temperature in the Central America example, used by the authors as comparison, was quite debated. The authors reference the paper by Ranero et al. (2008) who used BSR 1D modeling to infer the plate boundary temperature. However, a few years later, Harris and co-authors performed a different sets of models using the incoming plate heat flow measurements and, successively, ocean drilling results to infer quite different – and much lower – temperatures than what had been previously done by Ranero et al. For this specific paper it would be important to know what data have been used as input, what kind of modelling have been performed and the error bars. Depending on this, maybe also the discussion could be re-arranged.

The second point regards the role of fluids in guiding rupture nucleation and propagation. The authors refer to an overpressured plate boundary as unable to accumulate elastic strain (abstract and from line 131 onward) and therefore also unable to release it and they ultimately link the updip extent of seismic rupture to the hydrogeology of the plate boundary. However stress accumulation and release drive rupture nucleation. Co-seismic rupture propagation, instead, could be facilitated by low effective stress and fluids – see for example processes as fluid pressurization, just to cite one of them. In other words, the presence of fluids along a plate boundary might not be what is causing the rupture to arrest. In the ms by Ma

et al. it is not clear how rupture propagation and nucleation relate one another and the role of fluids in both processes.

The third point regards fluid quantification. The authors state that the sediments cannot be the source of fluids because of their scarcity. I would like to see some quantification. How much is "small volume" (line 113)? How much fluid can you extract – pore fluids and mineral dehydration? How much do you think might be the volume in the basement? How much do you need to permeate the plate boundary? Without quantification this whole discussion is just a speculation.

I hope these comments help.

Best Regards

Paola Vannucchi

Reviewer #2 (Remarks to the Author):

This manuscript presents a fairly straightforward and intriguing observation: in a grid of 2D seismic lines acquired over and adjacent to the rupture area of the 2014 Iquique earthquakes, the plate boundary exhibits prominent reflectivity outside the inferred rupture patch but does not exhibit such reflectivity inside the rupture patch. Taken on its face, that is a noteworthy correspondence that clearly could add to our understanding of what governs megathrust rupture, and as such may be a clearly significant contribution. The conceptual model the authors put forward is similarly simple: that areas of high reflectivity denote locations of elevated pore fluid pressure and/or enhanced fluid content along the fault which act to suppress the accumulation of elastic strain and therefore likelihood of rupture. They further observe that, comparing the spatial pattern of reflectivity to a pre-existing thermal model of the northern Chile margin, the high reflectivity zone extends down-dip to about where temperatures reach 12-150 °C, a regime often identified as the thermal condition for smectite clay dehydration to be complete. Smectite can yield a lot of water up to this temperature, which causes the excess fluid content in this concept. A further compelling observation is that the dip line to the north of the rupture area, MC04, shows high reflectivity all the way down in the un-slipped lateral equivalent of the 2014 slip patches.

The spatial correspondence of the observations is compelling. Assuming that the authors' assertion that the seismic data processing was done in a way that is consistent across all lines, then the observations themselves are very much worthy of prominent publication. I reviewed the supplementary information, and while the details of choices made to perform the pre-stack depth migration are a little scant, it seems fair to accept that this result (the spatial pattern of reflectivity) is not an artifact or manipulation.

The thermal structure of the profiles has been taken from a study referenced as Kellner)2007), which is an otherwise unpublished thesis (complete citation information is lacking, but I found it with an internet search). I briefly scanned through that work and it seems like a reasonable approach to thermal modeling, but I'm not expert in that field, and it doesn't seem to be calibrated or supported by seafloor heat flow data. Given all that, I would suggest some more detail be added (ideally a bit in the main text, and more in the supplement) on the thermal model (e.g., was the model with or without shear heating used and so on? What uncertainty is there in the thermal structure, especially with isotherms broadly parallel to the plate interface?).

I have concerns about some key parts of the interpretation of the observations, which I hope the authors could address in revision:

1. The model they present is that (A) reflective portions of the megathrust are fluid-rich and weak, do not accumulate elastic strain, and therefore did not slip in the 2014 earthquake, and (B) non-reflective portions are more effectively drained, therefore stronger, and did accumulate strain that led to the rupture and large slip. However, the seismic reflection lines were acquired in 2016, after the Iquique quake. So – can we really assume the non-reflective portions looked like that before April 2014? The fact that the line MC04 to the north is reflective through the full depth range would suggest to me that a logical explanation is that rupture modified the reflectivity (perhaps by draining the accumulated pore pressure?), rather than the other way around. In the authors' preferred reasoning, there is no explanation for why the area north of the rupture patch is reflective (in 2016), whereas this alternative explanation would. That is, which is the cause and which is the effect, when thinking about reflectivity and rupture area limits? To me, this is worth some consideration and perhaps a modification of the "Impact of fluid pressure" and "Implications"

sections.

2. The model also is a bit dated or oversimplified with regard to the concept of the up-dip limit. A great deal of work in the past decade or more has emphasized the conditional nature of slip mode, with transitional or temporally varying slow slip, tremor, locking, and dynamic effects overlapping in this region – for example as in Nankai, where all of those slip modes have been observed now in the shallow plate boundary where it is strongly reflective. Furthermore, dynamic rupture modeling and high speed friction experiments suggest slip can and does propagate into and through the “weak” shallow regions, depending on conditions. None of that is to say that the basic idea here is wrong – yes, the reflective zone might be indicative of a regime of velocity strengthening and therefore energy absorption that stops rupture propagation. I would recommend some attention to nuance in the discussion of the conceptual model to acknowledge that, rather than a simple “aseismic vs seismic” concept.

3. Finally, if there is room, I’d like to see a bit more rigorous analysis or at least discussion of the nature of the reflectivity. Reflectors require vertical velocity contrasts that might be due to pore fluids, but also might be due to other physical effects. For example, a hydrated and/or smectite-rich upper ocean basalt layer might evolve to higher seismic velocity through mineralogical change, rather than strictly pore pressure change. The studies that the pore fluid drainage model are based on are largely about compacting sediments, which may exhibit quite different rock physics than crustal rocks. Does the PSDM velocity model shed any light on the velocity values, thickness of units, or lateral gradients that yield the reflectors? Perhaps a full analysis can’t fit into a short publication like this, but some discussion could be included.

In summary, I'm intrigued by the results presented here and look forward to seeing them published. My suggestions above are intended to stimulate additional consideration of the possible causes of this very interesting dataset.

Reviewed by Harold Tobin

Manuscript No.: NCOMMS-21-33571-T

Title: Megathrust reflectivity reveals the updip limit of the 2014 Iquique earthquake rupture

Response to reviewers

We sincerely thank both reviewers for their constructive reviews, which helped us improve our manuscript. We carefully modified the main text, figures, and the supplementary information according to the suggestions from the reviewers. Please find below our answers for each comment in red text.

Kind regards,

Bo Ma on behalf of all co-authors

Answers to reviewer's comments

Referee 1: Paola Vannucchi

The ms presents seismic reflection data acquired offshore Northern Chile in the area of the 2014 Iquique EQ. The authors analyze the reflectivity of the plate boundary down-dip and along-strike and infer that the updip-propagation of the co-seismic rupture is related to the fluid content and possibly fluid overpressure along the plate boundary.

The ms is well written and the figures are appropriate and of good quality.

My review focuses on three points, that, in my opinion, need further thinking.

1. The first one regards the temperature proposed for the plate boundary at different depths. The authors mention numerical models performed by Kellner (2007). I reckon this is an unpublished thesis (Ms? PhD?).

Yes, it is an unpublished PhD thesis that is, however, accessible online. We added an in-depth discussion of different thermal models to the supplementary information. We estimated a new thermal model using an analytical approach, including friction heating, for comparison to the Kellner results. For the main manuscript we now use the newly derived thermal model.

An important reason why we had chosen the thermal model of Kellner (2007) in the original manuscript is that this model includes different shear stress values along the plate boundary. Compared to further south, the forearc heat flow in our study area (~21 °S) is higher than near 33°S, indicating a higher value of frictional heating than further south. The constraint Kellner uses to evaluate which frictional model is preferred is the cross-over between the depth at which megathrust seismogenic slip seems to stop and the temperature indicated by petrological results for this boundary (Figure 4.6 in the thesis). They conclude that a considerable amount of shear stress is needed for their model to be consistent with the seismological and petrological constraints. To further detect the effect of different shear stress on the thermal model, we

compared the curve of shear stress $\tau=67$ MPa (Suppl. Fig. 9, black dashed line) and $\tau=33$ MPa (Suppl. Fig. 9, orange dashed line) of Kellner (2007) and found that at the same depth under higher shear stress the thermal model has a higher temperature. In comparison, we also plot another thermal model without frictional heating of Cabrera et al. (2021), which is shown as the magenta dashed line in Suppl. Fig. 9. These comparisons indicate that the amount of shear heating has a huge impact on the predicted temperature at all depths.

The study of the up-dip limit at shallow depth (~15 km) and its correlation to the reflectivity pattern of the seismic data (Fig. 2; Suppl. Fig.8, grey horizon line) and the model with shear stress $\tau=67$ MPa show consistent features. The shear stresses resulting from the thermal models allow estimating the coefficients of basal friction μ_b using the static friction law $\tau = C + \mu_b \sigma_n$, where τ is the shear stress, C is the cohesion, and σ_n is the normal stress. Under shear stress $\tau=67$ MPa, the estimated average basal friction is $\mu_b=0.10$ (Kellner, 2007). This value is consistent with the global thermal measurement (0.03-0.13, Gao and Wang, 2014) and the orogeny force balance anticipation (Lamb, 2006).

We recognize that this discussion was missing in the original version and thus have added the following text to the supporting material:

Added text (lines 188-206 in Suppl.): An interesting feature is that the models for northern Chile show larger frictional heating compared to those observed in south-central Chile (Grevemeyer et al., 2003; Völker et al., 2011; Rotman and Spinelli, 2014). However, already Rotman and Spinelli (2014) suggested that frictional heating at the plate boundary increases northward, perhaps mimicking the increasing age of the subducting plate. Furthermore, patterns are consistent with heat flow anomalies over the marine forearc. Heat flow anomalies over the marine forearc are in the order of 50-60, 40-50, and 24-31 mW/m² at 39°S, 36°S and 33°S, respectively (Grevemeyer et al., 2003), decreasing northward and hence reflecting increasing crustal age of the subducting plate and supporting a decrease of basal heat flow. At 21°S, however, the age of the subducting plate has increased by roughly 20 Myr with respect to 33°S, but the forearc heat flow is in the order of 30-40 mW/m² (Springer and Forster, 1998) and thus higher than near 33°S, supporting higher values of frictional heating than found further south. It might be reasonable to hypothesize that sediment starved subduction erosion supports a higher degree of friction than the accretionary margin of south-central Chile, but this interpretation is beyond the scope of our work.

Both our analytical model and the numeric 2D model of Kellner (2007) show higher temperatures along the subduction megathrust fault with respect to other models for northern Chile. For example, Cabrera et al. (2021) (Suppl. Fig.8, magenta dashed line) did not consider any frictional heating with the argument that in south-central Chile shear heating was low and therefore they obtained lower temperatures.

Added figure (lines 302-308 in Suppl.): Suppl. Fig. 9: Comparison of different thermal models. Black dashed line: thermal model with shear stress $\tau=67$ MPa (Kellner, 2007); red dashed line: thermal model with frictional heating discussed in main text; orange dashed line: thermal model with shear stress $\tau=33$ MPa (Kellner, 2007); magenta dashed line: thermal model of ref.

(Cabrera et al. 2021); green dashed line: alternative thermal model computed without frictional heating. The approximate depth range of smectite clay dehydration is based on refs (Spinelli and Saffer, 2004; Saffer and Tobin, 2011; Kastner et al., 1991; Bekins et al., 1994; Underwood, 2007).

Temperature is a key point in this paper and I would like to see how this modelling is done.

We agree and thus made a special effort to expand the manuscript accordingly. We now show our own analytical thermal model with frictional heating by following the approach of England et al., (2018) to calculate the temperature as a function of depth at the interplate fault zone to (1) compare with Kellner's model; (2) update the thermal model we used in the main text. In our model, we used a frictional coefficient of $\mu=0.85$ and fluid pressure ratio (FPR) of 0.95 (McCaffrey et al., 2008). Frictional coefficient $\mu=0.85$ and FPR of 0.95 are the classical values for the oceanic and continental crust.

Our analytical model shows a similar thermal prediction as the 67 MPa model (black and red dashed lines in Suppl. Fig. 9), to some extent indicating higher temperature along the plate boundary fault with respect to other models. For example, another analytical model we built (Suppl. Fig. 9, green dashed line) and the model of Cabrera et al. (2021) (Suppl. Fig. 9, magenta dashed line) did not consider any frictional heating and therefore obtained lower temperatures. Kellner's model with a lower shear stress $\tau=33$ MPa takes partial shear stress into account so that the temperature is higher than models without shear stress but lower than the 67 MPa model and our new model with frictional heating. The prediction depth of both Kellner's $\tau=67$ MPa model (< 20 km) and our analytical model (< 20 km) are close to the reflectivity termination depth (~ 15 km) observed in the MCS images at the upper threshold of the clay dehydration temperature of 150°C (Suppl. Fig. 9).

Integrating the higher frictional heating in the Northern Chile margin and the matching of the reflectivity pattern with both our analytical model and Kellner's 67 MPa model, we believe both our new model with frictional heating and Kellner's 67 MPa is reasonable in Northern Chile. We added the following information to the supporting material:

Added text (lines 98-174 in Suppl.): It has long been recognized that the rupture zone of subduction zone megathrust earthquakes is at least partially controlled by the thermal state of the fault zone (Tichelaar and Ruff, 1993; Hyndman and Wang, 1993). Analytical models revealed that the geometry of the subduction zone, the thermal state of the incoming subducting plate and the shear or frictional heating along the megathrust are critical parameters controlling megathrust temperatures (Molnar and England, 1990; England, 2018), which in turn, define the seaward and landward limit of large subduction earthquakes rupture zones. The seaward or updip limit is generally associated with temperatures of $100^\circ\text{-}150^\circ\text{C}$, marking the smectite to chlorite transition (Hyndman and Wang, 1993) or a suite of diagenetic reactions and release of water from underthrust sediments (Spinelli and Saffer, 2004) and/or basement (Kameda et al., 2011). The landward or downdip limit is assumed to be associated with a critical temperature of $350^\circ\text{-}400^\circ\text{C}$, which marks the transition from stick-slip to stable sliding at the onset of quartz

and feldspar plasticity of continental crustal rocks (Scholz, 1988).

The geometry of the subduction zone is readily known from geophysical data or the hypocentral depth of large megathrust earthquakes and the basal heat flow is defined by the age of the incoming oceanic plate. Most thermal parameters of subduction zones show little variation along the Pacific Ring of Fire and are well established, especially for Chile (Völker et al., 2011; Rotman and Spinelli, 2014). Based on these data, most subduction zones have been studied using two-dimensional thermal models. Here, we use two different approaches. First, we calculate the thermal state along the plate interface or megathrust fault using analytical solutions based on the formalism of ref. (England, 2018) and second, we consider a 2-dimensional thermal model of ref. (Kellner, 2007) for northern Chile, incorporating corner flow in the mantle wedge.

Analytical expressions, which relate surface heat flux to temperature, geometrical constraints, and shear stress, provide an efficient approach to study the thermal state of the megathrust fault and are discussed in detail by ref. (Molnar and England, 1990; Molnar and England, 1995; England, 2018). We follow the approach of ref. (England, 2018) and first calculate the temperature T_f as a function of the depth z on the interplate fault zone as:

$$(1) T(z) = K_m T_0 z / SK_s [\pi \kappa (t_0 + t_s)]^{1/2}$$

where $S(z) = 1 + bK_m[(V_n z \sin \delta) / \kappa]^{1/2} / K_s$. K_m ($3.3 \text{ Wm}^{-1}\text{K}^{-1}$) and K_s ($2.55 \text{ Wm}^{-1}\text{K}^{-1}$) are the mantle and forearc thermal conductivity, respectively. T_0 is the asthenospheric mantle temperature (1300°C) and κ is the thermal diffusivity ($10^{-6}\text{m}^2\text{s}^{-1}$). t_0 is the average age of the subducting plate (50 Myr; Muller et al., 2008), whereas t_s is the time it takes the lithosphere to subduct to a depth z . V_n is the convergence rate normal to the subduction ($\sim 70 \text{ km/Myr}$), δ is the dip angle of subduction, and $b(\pi^{-1/2})$ is a factor that depends on the specific geometry (Molnar and England, 1995).

The dip angle of subduction was taken directly from the seismic reflection images (Fig. 2), and the time t_s is computed by dividing the integrated downdip length of the fault surface by V_n (McCaffrey et al., 2008). We neglect the effect of the horizontal heat flow. To calculate the radiogenic heat production T_r in the forearc crust we used the following expression (McCaffrey et al., 2008):

$$(2) T_r(z) = A_r z^2 / (2K_s S(z))$$

where A_r is the radiogenic heat production rate (10^{-6} Wm^{-3}) (Grevemeyer et al., 2003). Radiogenic heat production adds $0\text{-}45^\circ\text{C}$ to the fault temperature from the trench axis up to the downdip limit.

Further, we include in our model frictional shear heating $T_{sh}(z)$ on the thermal field by using:

$$(3) T_{sh}(z) = \tau(z) V_t z / (K_s S(z))$$

where $\tau(z)$ is the shear stress on the fault and V_t is the total slip rate (England, 2018; Molnar and England, 1995). $\tau(z)$ on a gently dipping fault at shallow depth is approximately

$$(4) \tau(z) = \mu(\sigma_n(z) - p(z))$$

where μ is the friction coefficient, σ_n is the normal stress applied on the fault plane (approximately the overburden pressure), and p is the pore fluid pressure. Following ref. (McCaffrey et al., 2008), we use $\mu = 0.85$, $\sigma_n(z) = \rho g z$ and $p(z) = 0.95 \sigma_n(z)$ with acceleration of gravity $g = 9.8 \text{ ms}^{-2}$ and the average crustal density $\rho = 2500 \text{ kgm}^{-3}$.

The final predicted temperature on the fault plate boundary $T_f(z)$ is the sum of Eqs. (1)-(3)

(i.e., $T_f(z) = T(z) + T_r(z) + T_{sh}(z)$). Fig. 2 shows the estimated temperature values for $T_f(z)$ along our seismic reflection lines.

We compare our model to the numerical model of ref. (Kellner, 2007). The geometry of this model is based on a suite of geophysical data (Oncken, 2003) and thermal parameters were rated against a number of observed features, including the maximum depth of subduction thrust earthquakes and observed heat flow. Interestingly, the maximum depth of seismic faulting of megathrust earthquakes in northern Chile occurs at 40-50 km (Tichelaar and Ruff, 1991; Peyrat et al., 2010), suggesting that temperatures of 350°-400°C are reached at ~40-50 km, too. To mimic this feature, ref. (Kellner, 2007) had to introduce a considerable amount of shear heating, in the order of $\tau=33$ MPa to $\tau=67$ MPa, with the upper limit providing a better fit to the data. The predictions from the $\tau=67$ MPa model mimic the prediction of our preferred analytic solution down to a depth of approx. 30 km. At greater depth, the models differ with the ref. (Kellner, 2007) model showing somewhat lower temperatures. The observed differences may stem from the effects of the corner flow incorporated into the numerical model and a change in dip angle, which is not considered in the analytic model. We also compare our model with other thermal models from the Northern Chilean margin. A comparison of all the models for the temperature along the plate interface is shown in Suppl. Fig. 9, including the thermal model with shear stress $\tau=67$ MPa (Kellner, 2007) as the black dashed line, the thermal model with frictional heating we used in the main text as the red dashed line, the thermal model with shear stress $\tau=33$ MPa (Kellner, 2007) as the orange dashed line, the thermal model of ref. (Cabrera et al., 2021) as the magenta dashed line, and a thermal model we built without the frictional heating as the green dashed line. Please note that for the study of the updip limit at shallow depth (~15 km) and its correlation to the reflectivity pattern of the seismic data, the model we established with frictional heating and the model with shear stress $\tau=67$ MPa of ref. (Kellner, 2007) show consistent features.

Temperature in the Central America example, used by the authors as comparison, was quite debated. The authors reference the paper by Ranero et al. (2008) who used BSR 1D modeling to infer the plate boundary temperature. However, a few years later, Harris and co-authors performed a different sets of models using the incoming plate heat flow measurements and, successively, ocean drilling results to infer quite different – and much lower – temperatures than what had been previously done by Ranero et al. For this specific paper it would be important to know what data have been used as input, what kind of modelling have been performed and the error bars. Depending on this, maybe also the discussion could be re-arranged.

We agree that different thermal model setups may lead to varying thermal fields. Actually, our new analytical thermal model is also lower than the 67 MPa model (Kellner, 2007, Suppl. Fig. 9 black and red dashed lines). Although differences in thermal fields exist, the gap in depth reached at the peak temperature of smectite clay dewatering (~150°C) is not significant. Moreover, with varying effective coefficient of basal friction μ_b values from 0.03-0.13 (Suppl. Fig. 10), which is consistent with the global thermal measurement (Gao and Wang, 2014), the results show that the depth gap between reflectivity and thermal model at 150°C is less than 5 km and with higher μ_b the thermal model and reflectivity have a better spatial matching relationship.

Added text (lines 176-186 in Suppl.): As previously mentioned in the main text, the established new thermal model uses a friction coefficient $\mu=0.85$ and pore fluid pressure $\lambda=0.95$ (McCaffrey et al., 2008). Since these values vary in each tectonic setting, we applied different μ_b in the new analytical thermal model in Suppl. Fig. 10. The effective coefficient of basal friction μ_b depends on both the friction coefficient μ and pore fluid pressure λ along the fault zone: $\mu_b = \mu(1-\lambda)$ (Hubbert and Rubey, 1959). Based on this formula, the μ_b of the model in our main text is 0.0425, which is shown as the red dashed line. We applied a range of μ_b from 0.03-0.13, consistent with the global thermal measurement (Gao and Wang, 2014). In this range, the predictions depth of the analytical model (< 20 km) is close to the downdip limit of the megathrust reflectivity (~15 km) observed from MCS images at the upper threshold of the clay dehydration temperature of 150°C (Suppl. Fig. 10). Moreover, the thermal model and reflectivity show a better spatial matching with a higher effective coefficient of basal friction μ_b value.

Added figure (lines 309-318 in Suppl.): Suppl. Fig. 10: New thermal model with varying effective coefficient of basal friction μ_b values. The range of μ_b is from 0.03-0.13, consistent with the global thermal measurement (Gao and Wang, 2014). The estimated average basal friction $\mu_b=0.1$ of Kellner 67 MPa is shown as a black dashed line. Using the same basal friction value, our new analytical model is shown as a magenta dashed line. In the main text, we used a friction coefficient of $\mu=0.85$ and pore fluid pressure $\lambda=0.95$ (McCaffrey et al., 2008). The effective coefficient of basal friction μ_b depends on both the friction coefficient μ and pore fluid pressure λ along the fault zone: $\mu_b = \mu(1-\lambda)$ (Hubbert and Rubey, 1959). Based on this formula, the μ_b of the model in our main text is 0.0425, which is shown as red dashed line. $\mu_b=0.03$ and $\mu_b=0.13$ are shown as green and blue dashed lines, respectively.

2. The second point regards the role of fluids in guiding rupture nucleation and propagation. The authors refer to an overpressured plate boundary as unable to accumulate elastic strain (abstract and from line 131 onward) and therefore also unable to release it and they ultimately link the updip extent of seismic rupture to the hydrogeology of the plate boundary. However stress accumulation and release drive rupture nucleation. Co-seismic rupture propagation, instead, could be facilitated by low effective stress and fluids – see for example processes as fluid pressurization, just to cite one of them. In other words, the presence of fluids along a plate boundary might not be what is causing the rupture to arrest. In the ms by Ma et al. it is not clear how rupture propagation and nucleation relate one another and the role of fluids in both processes.

We agree with the reviewer that our strain accumulation versus strain release model was oversimplified and not sufficiently reflecting the knowledge that had been gained (especially in recent years) on different slip characteristics along the shallow plate boundary. This was also a main concern of reviewer 2 (comment #2). We also realize that the rupture nucleation versus rupture propagation aspect was not sufficiently covered. In the revised version we now explicitly state that while the shallow plate-boundary of Northernmost Chile may not nucleate a giant megathrust earthquake, slip may well propagate into this weak and overpressured region.

To account for these considerations, we modified some sections of the introduction and the discussion.

Modified text (lines 36-38 in the main text): Pore fluid pressure in excess of hydrostatic pressure diminishes fault strength (Hubbert and Rubey, 1959) and enables a wide spectrum of transient, predominantly slow, earthquake phenomena especially along shallow subduction zone plate boundaries (Saffer and Wallace, 2015).

Modified text (lines 130-137 in the main text): In many subduction zones, some of which have been targeted by scientific ocean drilling (Northern Barbados, Costa Rica, Nankai Trough, Hikurangi), such coherent, high reflectivity corresponds to the shallow region of the plate boundary that does not nucleate giant earthquakes (Ranero et al., 2008; Shipley et al., 1994; Moore et al., 1998; Bangs et al., 1999; Spinelli and Wang, 2008; Tobin and Saffer, 2009; Bell et al., 2010; Bangs et al., 2015). Advances in seafloor geodesy, in concert with scientific drilling of shallow plate boundaries and numerical modelling, have further proven that these regions can host a wide spectrum of slow earthquake phenomena with event durations up to some years (Saffer and Wallace, 2015; Araki et al., 2017). For example, on the Nankai margin, recurring slow-slip events along the high-reflective shallow plate boundary, accommodate 30-55% of the plate convergence (Araki et al., 2017).

Added text (lines 191-209 in the main text): Furthermore, the high fluid pressures and the low effective stresses may promote strain release during slow and aseismic events that occur at shorter intervals compared to large megathrust earthquakes. While observed along the Nankai (Araki et al., 2017), Hikurangi (Wallace et al., 2016), and Costa Rica (Davis et al., 2015) margins, such slow and shallow earthquake phenomena have not been resolved off Northern Chile, where Global Navigation Satellite System-Acoustic (GNSS-A) seafloor observations and borehole observatories are lacking.

Modified text (lines 258-262 in the main text): Models using only land geodetic measurements, however, cannot resolve whether the shallowest part of a plate boundary is locked over a time period that is long enough to accumulate sufficient elastic energy to nucleate a large earthquake (Metois and Socquet, 2016; Almeida et al., 2018; Kosari et al., 2020; Lindsey et al., 2021). Alternatively, the shallow plate boundary may also creep at plate convergence rates or release energy during frequent slow-slip events, low-frequency earthquakes, or episodic tremor and slip (Saffer and Wallace, 2015; Yokota and Ishikawa, 2020).

Added text (lines 265-267 in the main text): This does not exclude the possibility that earthquake rupture arising from the downdip seismogenic zone may propagate into and through the shallow plate boundary, as it has been observed and inferred from other margins (Vannucchi et al., 2017; Kodaira et al., 2012; Maksymowicz et al., 2017).

Modified text (lines 273-276 in the main text): Seismic reflection data, which are sensitive to fluid-pressure variations, can help to identify the shallow, velocity strengthening part of the plate boundary that may not nucleate a large earthquake and that may accumulate and

release strain in a different manner and at different timescales (intervals), compared to the non-reflective seismogenic zone located further downdip.

3. The third point regards fluid quantification. The authors state that the sediments cannot be the source of fluids because of their scarcity. I would like to see some quantification. How much is “small volume” (line 113)? How much fluid can you extract – pore fluids and mineral dehydration? How much do you think might be the volume in the basement? How much do you need to permeate the plate boundary? Without quantification this whole discussion is just a speculation.

We agree that there is uncertainty with respect to fluid quantities. We calibrated the reflection amplitudes and estimated the reflectivity strength (see answer to Reviewer 2, comment #3). We also add a reference to a new paper that presents a velocity model for the incoming plate and shows anomalously low velocities in the upper 5 km of the oceanic crust when compared to “normal” Nazca plate crust, providing evidence for hydration and/or fluid infiltration via cracks of the crust.

The oceanic basement is covered by thin sediments and has never been drilled in northern Chile. We extend the discussion on water volume by quantifying the thickness of the oceanic sediment off Northern Chile and by using information on sediment compositions from other margins such as Costa Rica. We measured sediment thickness on the oceanic plate along the seismic profiles. Sediment thickness is around 100-200 m (Suppl. Fig.7, red dots of error bar and yellow solid lines in (a)-(d)).

We modified the following passages of the discussion:

Added text (lines 144-168 in the main text): The small volume of trench fill along the Northern Chilean margin (Fig. 2 and Suppl. Fig. 8, also compare to Geersen et al., 2018), however, makes it questionable whether subducting sediments are a significant source of fluids. If smectite clay accounts for 50% of the bulk sediment, as observed in other subduction zones with typical deep-water pelagic and hemipelagic environments of mudstones such as the Nankai Trough, Cascadia, Barbados Ridge, Costa Rica (Underwood, 2007; Vannucchi et al., 2017), this would correspond to 8-10 wt% of water (~15-20 vol%). Considering that the thickness of sediments that rest on the igneous oceanic basement is generally less than 200 m (and often even less than 100 m) along the Northern Chilean margin (Suppl. Fig. 8, red dots of error bar and yellow solid lines in (a)-(d)), water release from mineral dehydration within the subducting sediment must be quite limited, even when considering high smectite clay concentrations.

Added text (lines 184-191 in the main text): Although the incoming oceanic crust off northern Chile has not been drilled, recent seismic data suggest that the upper crustal velocity of the incoming plate beneath the outer rise is anomalously low, suggesting presence of mineral hydration and/or pervasive fluid-filled cracks to a depth of ~5 km (Myers, et al., doi: 10.1029/2021JB023169 (2022)). We conclude that the coherent highly reflective shallow plate boundary off Northern Chile, which behaved aseismic during the 2014 Iquique earthquake, is

fluid-rich, and that the source of the fluid is intergranular and fracture porosity in seismic layer 2A (Anderson et al., 1976; Jarrard, 2003; Kameda et al., 2011) of the oceanic crust and mineral bound water released through dehydration of the weathered clay-bearing oceanic basalt. This may promote reduced coupling as suggested by Moreno et al., (2014) for the southern boundary of the 2010 Maule earthquake.

Modified text (lines 180-183 in the main text): To calculate the total amount of water that could be liberated from the scarce subducting sediments and the subducting weathered oceanic basement, one would need to know the exact composition of the sediments as well as the depth extent and degree of weathering of the clay-bearing basalts.

Added figure (lines 291-300 in Suppl.): Suppl. Fig. 8: Oceanic crust of pre-stack depth migrated section along seismic dip-lines. The error bar on top indicates the sediment upon the oceanic crust, in which the red dots show the average thickness of sediment on oceanic crust. The upper limit of the error bar represents the maximum thickness of sediments, while the lower limit indicates the minimum thickness of sediment. The maximum thickness anomaly along MC06 is due to more sediment in the trench than along the other seismic lines. In (a)-(d), yellow solid lines indicate the thickness of sediment on oceanic crust. Due to the bending of the oceanic crust, the sediments are accumulated in half-graben structures. (a): seismic line MC04; (b): seismic line 06; (c): seismic line 25; (d): seismic line 17. Vertical exaggeration is 3.

I hope these comments help.

Best Regards

Paola Vannucchi

Referee 2: Harold Tobin

This manuscript presents a fairly straightforward and intriguing observation: in a grid of 2D seismic lines acquired over and adjacent to the rupture area of the 2014 Iquique earthquakes, the plate boundary exhibits prominent reflectivity outside the inferred rupture patch but does not exhibit such reflectivity inside the rupture patch. Taken on its face, that is a noteworthy correspondence that clearly could add to our understanding of what governs megathrust rupture, and as such may be a clearly significant contribution. The conceptual model the authors put forward is similarly simple: that areas of high reflectivity denote locations of elevated pore fluid pressure and/or enhanced fluid content along the fault which act to suppress the accumulation of elastic strain and therefore likelihood of rupture. They further observe that, comparing the spatial pattern of reflectivity to a pre-existing thermal model of the northern Chile margin, the high reflectivity zone extends down-dip to about where temperatures reach 12-150 °C, a regime often identified as the thermal condition for smectite clay dehydration to be complete. Smectite can yield a lot of water up to this temperature, which causes the excess fluid content in this concept. A further compelling observation is that the dip line to the north of the rupture area, MC04, shows high reflectivity all the way down in the unslipped lateral equivalent of the 2014 slip patches.

The spatial correspondence of the observations is compelling. Assuming that the authors' assertion that the seismic data processing was done in a way that is consistent across all lines, then the observations themselves are very much worthy of prominent publication. I reviewed the supplementary information, and while the details of choices made to perform the pre-stack depth migration are a little scant, it seems fair to accept that this result (the spatial pattern of reflectivity) is not an artifact or manipulation.

The thermal structure of the profiles has been taken from a study referenced as Kellner (2007), which is an otherwise unpublished thesis (complete citation information is lacking, but I found it with an internet search). I briefly scanned through that work and it seems like a reasonable approach to thermal modeling, but I'm not expert in that field, and it doesn't seem to be calibrated or supported by seafloor heat flow data. Given all that, I would suggest some more detail be added (ideally a bit in the main text, and more in the supplement) on the thermal model (e.g., was the model with or without shear heating used and so on? What uncertainty is there in the thermal structure, especially with isotherms broadly parallel to the plate interface?).

This comment raises very similar concerns to the first reviewer. We take this remark very seriously and have adjusted the text accordingly. Please refer to Comment #1 of Reviewer 1 for details on how we adjusted the aspects of temperature in the main text and in the Supplementary information.

I have concerns about some key parts of the interpretation of the observations, which I hope the authors could address in revision:

1. The model they present is that (A) reflective portions of the megathrust are fluid-rich and weak, do not accumulate elastic strain, and therefore did not slip in the 2014 earthquake, and (B) non-reflective portions are more effectively drained, therefore stronger, and did accumulate strain that led to the rupture and large slip. However, the seismic reflection lines were acquired in 2016, after the Iquique quake. So – can we really assume the non-reflective portions looked like that before April 2014? The fact that the line MC04 to the north is reflective through the full depth range would suggest to me that a logical explanation is that rupture modified the reflectivity (perhaps by draining the accumulated pore pressure?), rather than the other way around. In the authors' preferred reasoning, there is no explanation for why the area north of the rupture patch is reflective (in 2016), whereas this alternative explanation would. That is, which is the cause and which is the effect, when thinking about reflectivity and rupture area limits? To me, this is worth some consideration and perhaps a modification of the "Impact of fluid pressure" and "Implications" sections.

After collection of the post-earthquake seismic reflection data with *RV Langseth*, we tried to identify transient changes in reflectivity across the 2014 earthquake by comparing the seismic images to the 1995 *RV SONNE* data. In a nutshell, we do not observe any changes in the shallow high-reflective section of the subduction zone that did not rupture. Further down-dip, i.e. at seismogenic zone depths, we do not find any evidence for high reflectivity in the pre-earthquake data. However, we cannot ultimately exclude that this is due to resolution issues of the older data which have been collected in 1995 with a significantly shorter streamer.

We agree that the transient change aspect was not well covered in the manuscript and we now

refer to it in the revised version of the Impact of fluid pressure” and “Implications” sections.

Modified text (lines 226-254 in the main text): For the Antofagasta Mw 8.0 earthquake, Husen and Kissling (2001) compared upper plate seismic velocities from before and after the earthquake and suggested that seismic rupture enabled fluids to migrate upwards from the plate boundary into the upper plate. Such a process could reduce plate boundary reflectivity during the early postseismic phase and would thus provide a possible explanation for the observed low reflectivity within the 2014 Iquique rupture area. In general, there is little information on transient changes of reflectivity during a seismic cycle. Prior to the 2014 Iquique earthquake ref. (Storch et al., 2021) observed high plate boundary reflectivity down to a depth of 16 km. This indicates that the shallow plate boundary remains fluid rich throughout the seismic cycle. Since plate boundary reflectivity at the depths of the seismogenic zone is not resolved prior to the 2014 Iquique earthquake (Storch et al., 2021), we cannot conclude whether it was altered by the 2014 Iquique earthquake. However, as the updip end of seismicity, which correlates to the updip extent of the 2014 Iquique rupture, is stable in space throughout the seismic cycle, as also observed for the Sumatran (Tilmann et al., 2010) and the South Chilean margin (Lange et al., 2018), we presume that the reflectivity may also not change significantly over time. Although the ultimate reason for the higher plate boundary fluid content at depths beyond 15 km to the north of the 2014 Iquique earthquake remains enigmatic, our seismic reflection data clearly suggest a hydrogeological control on the updip extent and likely also the along-strike extent of seismic rupture along the erosive Northern Chilean continental margin.

2. The model also is a bit dated or oversimplified with regard to the concept of the up-dip limit. A great deal of work in the past decade or more has emphasized the conditional nature of slip mode, with transitional or temporally varying slow slip, tremor, locking, and dynamic effects overlapping in this region – for example as in Nankai, where all of those slip modes have been observed now in the shallow plate boundary where it is strongly reflective. Furthermore, dynamic rupture modeling and high speed friction experiments suggest slip can and does propagate into and through the “weak” shallow regions, depending on conditions. None of that is to say that the basic idea here is wrong – yes, the reflective zone might be indicative of a regime of velocity strengthening and therefore energy absorption that stops rupture propagation. I would recommend some attention to nuance in the discussion of the conceptual model to acknowledge that, rather than a simple “aseismic vs seismic” concept.

This comment touches similar aspects as comment #2 from the first reviewer. Our discussion was not sufficiently reflecting the gained knowledge on rupture styles and slip phenomena along shallow plate boundaries. We carefully reviewed the rupture nucleation versus rupture propagation and the seismic versus aseismic (and strain accumulation versus strain release) models. We now discuss our observations in the context of what had been learned on the transient behavior of the shallow most plate boundary elsewhere (Nankai, Hikurangi, Costa Rica).

To account for these considerations, we modified the following sections of the discussion.

Modified text (lines 36-38 in the main text): Pore fluid pressure in excess of hydrostatic pressure

diminishes fault strength (Hubbert and Rubey, 1959) and enables a wide spectrum of transient, predominantly slow, earthquake phenomena especially along shallow subduction zone plate boundaries (Saffer and Wallace, 2015).

Modified text (lines 130-137 in the main text): In many subduction zones, some of which have been targeted by scientific ocean drilling (Northern Barbados, Costa Rica, Nankai Trough, Hikurangi), such coherent, high reflectivity corresponds to the shallow region of the plate boundary that does not nucleate giant earthquakes (Ranero et al., 2008; Shipley et al., 1994; Moore et al., 1998; Bangs et al., 1999; Spinelli and Wang, 2008; Tobin and Saffer, 2009; Bell et al., 2010; Bangs et al., 2015). Advances in seafloor geodesy, in concert with scientific drilling of shallow plate boundaries and numerical modelling, have further proven that these regions can host a wide spectrum of slow earthquake phenomena with event durations up to some years (Saffer and Wallace, 2015; Araki et al., 2017). For example, on the Nankai margin, recurring slow-slip events along the high-reflective shallow plate boundary, accommodate 30-55% of the plate convergence (Araki et al., 2017).

Added text (lines 191-209 in the main text): Furthermore, the high fluid pressures and the low effective stresses may promote strain release during slow and aseismic events that occur at shorter intervals compared to large megathrust earthquakes. While observed along the Nankai (Araki et al., 2017), Hikurangi (Wallace et al., 2016), and Costa Rica (Davis et al., 2015) margins, such slow and shallow earthquake phenomena have not been resolved off Northern Chile, where Global Navigation Satellite System-Acoustic (GNSS-A) seafloor observations and borehole observatories are lacking.

Modified text (lines 258-262 in the main text): Models using only land geodetic measurements, however, cannot resolve whether the shallowest part of a plate boundary is locked over a time period that is long enough to accumulate sufficient elastic energy to nucleate a large earthquake (Metois and Socquet, 2016; Almeida et al., 2018; Kosari et al., 2020; Lindsey et al., 2021). Alternatively, the shallow plate boundary may also creep at plate convergence rates or release energy during frequent slow-slip events, low-frequency earthquakes, or episodic tremor and slip (Saffer and Wallace, 2015; Yokota and Ishikawa, 2020).

Added text (lines 265-267 in the main text): This does not exclude the possibility that earthquake rupture arising from the downdip seismogenic zone may propagate into and through the shallow plate boundary, as it has been observed and inferred from other margins (Vannucchi et al., 2017; Kodaira et al., 2012; Maksymowicz et al., 2017).

Modified text (lines 273-276 in the main text): Seismic reflection data, which are sensitive to fluid-pressure variations, can help to identify the shallow, velocity strengthening part of the plate boundary that may not nucleate a large earthquake and that may accumulate and release strain in a different manner and at different timescales (intervals), compared to the non-reflective seismogenic zone located further downdip.

3. Finally, if there is room, I'd like to see a bit more rigorous analysis or at least discussion of the nature of the reflectivity. Reflectors require vertical velocity contrasts that might be due to pore fluids, but also might be due to other physical effects. For example, a hydrated and/or smectite-rich upper ocean basalt layer might evolve to higher seismic velocity through mineralogical change, rather than strictly pore pressure change.

The studies that the pore fluid drainage model are based on are largely about compacting sediments, which may exhibit quite different rock physics than crustal rocks. Does the PSDM velocity model shed any light on the velocity values, thickness of units, or lateral gradients that yield the reflectors? Perhaps a full analysis can't fit into a short publication like this, but some discussion could be included.

To give a more quantitative view of the reflector amplitudes of the seismic sections we calibrated the amplitudes and estimated the reflectivity strength (envelope of the reflectivity). We included the new Supplementary Fig. 7 with a colour-coded reflection strength. There is a clear separation of values (colours) of the upper plate reflection strength and values along the plate boundary.

We agree that the v_p model may show the lateral gradients that yield the reflectors. However, due to the thin sediments and strong seafloor multiples along our profiles, we decided to choose an initial v_p model based on the OBS 3D v_p model of the refraction seismic data from the *RV Langseth*. Referring to the values of the OBS velocity model for the shallow depth, we give a gradient starting from the seafloor and build a velocity model without lateral variations for the pre-stack depth migration (see Suppl. Fig. 5). Although the incorrect v_p model may affect the depth of the layers (different velocity models may result in 1-2 km in-depth), in this manuscript, we only focus on the spatial distributions of reflectivity along the interface and the rupture. Therefore, small changes in depth do not affect our observations.

Added text (lines 88-95 in Suppl.): To quantify the amplitudes of the seismic sections, the reflection coefficient was estimated based on the ratio of the seafloor reflection to the seafloor multiple reflection. Due to interference of several reflector elements resulting in inverse and mix phased signals especially in the crustal overburden the envelope was calculated representing the absolute reflection strength (Suppl. Fig. 7). The plate boundary shows a unique reflection strength of 0.005 to 0.0075 in a depth range between 15 to 35 km, whereas the internal crustal reflector elements show a higher variability ranging from 0.0075 to 0.025 at a depth range of 5 to 15 km.

Added figure (lines 280-283 in Suppl.): Suppl. Fig. 7: Estimated reflection strength of pre-stack depth migrated seismic image of seismic line MC04. The plate boundary shows a unique reflection strength than the internal crustal reflector elements. Vertical exaggeration is 1.

In summary, I'm intrigued by the results presented here and look forward to seeing them published. My suggestions above are intended to stimulate additional consideration of the possible causes of this very interesting dataset.

Reviewed by Harold Tobin

REVIEWERS' COMMENTS

Reviewer #1 (Remarks to the Author):

I am happy with how the authors reply to my comments. I have no further comments.

Best Regards

Paola Vannucchi